# Sustainable Applications for Utilizing Antimony Tailing Coarse Aggregate (ATCA) in Concrete: Characteristic of ATCA and Toxicity Risks of Concrete

**DOI:** 10.3390/ma14195480

**Published:** 2021-09-22

**Authors:** Jianqun Wang, Long Li, Longwei Zhang, Bei Li, Renjian Deng, Defeng Shi

**Affiliations:** 1Hunan Provincial Key Laboratory of Structures for Wind Resistance and Vibration Control, School of Civil Engineering, Hunan University of Science and Technology, Xiangtan 411201, China; lilong15173271886@163.com (L.L.); bei620bei@163.com (B.L.); 800912deng@sina.com (R.D.); 2School of Resources, Environment and Safety Engineering, Hunan University of Science and Technology, Xiangtan 411201, China; 1020132@hnust.edu.cn

**Keywords:** antimony tailing waste rock (ATWR), antimony tailing coarse aggregate (ATCA), natural coarse aggregate (NCA), chemical composition, mechanical properties, quality of coarse aggregate (CA), leaching test

## Abstract

In this research, the sustainable applications for utilizing antimony tailing coarse aggregate (ATCA) in concrete is investigated. Comprehensive verifications were performed by a series of experiments on the characteristic of ATCA and the toxicity risks of concrete. Firstly, a real case study of utilization of ATCA as a complete substitute for natural coarse aggregate (NCA) in high strength concrete was conducted. Then, chemical composition of ATCA was tested. It is demonstrated that the essential mineral is SiO_2_ and the lithology of ATCA is quartzite. The mechanical properties, coarse quality of ATCA, and NCA were studied and compared. The compressive strength, splitting tensile strength, and compressive elastic modulus of ATWR are 221.51 MPa, 5.93 MPa, and 3.33 × 10^4^ MPa, which are 1.31, 2.22, 1.40 times of that of NR, respectively. All of the quality control indices of ATCA meet the requirements of the current industry standards of China. Finally, the toxicity risks of ATCA concrete were investigated. It is illustrated that the leaching of main heavy metals including Sb, As, Hg, Pb, Cd, and Zn in the ATCA concrete under different pH conditions are below the regulatory limits. The utilization of antimony tailing has significant environmental and economic benefits.

## 1. Introduction

Antimony (Sb) is a metalloid element belonging to Group 15 of the periodic table, naturally occurring in the Earth’s crust [1]. Sb is usually alloyed with other kind of metals and widely used in different industrial fields [2,3]. Due to the excellent metallogenic environment to form large and super large deposits, China is the country with the largest reserves of antimony mineral resources [4,5]. The Xikuangshan (XKS) Sb mine, located near Lengshuijiang City, Hunan Province, China, is the largest Sb mine in the world. Large-scale mining and smelting activities have occurred for over 120 years in this area, where it has for a long time been famously named the ‘‘World Antimony Capital’’ [6,7,8].

However, Sb and its compounds are considered to be hazardous to human health, causing degradation in liver and low blood sugar [9,10]. In the processes of mining and smelting, large quantities of Sb are released to the surrounding environment [11,12,13]. The mined antimony ore is generally broken into crushed stones with a certain particle size and used for artificial beneficiation. Plenty of low-grade antimony ore is selected and abandoned as antimony tailing waste rock (ATWR) [14]. Some of the ATWRs are used for backfilling, and the others are stored in a tailing dam, or abandoned in the open air nearby the mining area [15,16]. As illustrated in Figure 1, during the field investigation, the authors found that there were millions of tons of ATWRs in XKS. These ATWRs not only occupy valuable land resources but also induce significant pollution to the environment [17,18].

In recent years, the green and sustainable production of concrete receives much attention [19,20,21]. Aggregate continues to be increased as more concrete is consumed all over the world [22,23]. Since stone and sand are non-renewable resources, the traditional ways to produce aggregate is not sustainable [24,25]. Utilizing ore tailings as valuable concrete raw materials is a feasible treating method, which has received increasing attention for the past years [26,27,28]. Gayana and Chandar proposed that some waste materials, such as iron ore tailings, recycled concrete aggregate, glass, plastics, and quartz sandstone, could be utilized as replacement for coarse aggregates in concrete [29]. Zhao et al. investigated the utilization of iron ore tailings as fine aggregate in ultra-high performance concrete, the tailings were characterized by techniques like SEM, XRD, and nanoindentation [30]. Lv et al. reported that the iron tailings could be utilized as a complete replacement of normal aggregates in dam concrete. It was demonstrated that iron tailing concrete had a higher specific gravity and water consumption than that of natural aggregate (NA) concrete due to the different characteristic of iron tailing aggregate [31]. Vijayaraghavan et al. investigated the effect of using alternatives for both fine and coarse aggregates with copper slag, iron slag, and recycled concrete aggregate with various proportions of mix. Based on the characteristics of coarse aggregates and the mechanical properties of the concrete samples, they found that 40% of copper slag, 40% iron slag, and 25% of recycled concrete aggregate possessed more strength than the conventional concrete mix [32]. Thus, it can be concluded that the performance concrete is affected by the characteristic of tailing aggregate. We proposed utilizing antimony tailing coarse aggregate (ATCA) as a complete substitute for natural coarse aggregate (NCA) in high strength concrete in previous work, providing a new way for utilizing ATWRs [33], while the characteristic of ATCA has not been reported.

Besides, the toxicity risks of heavy metals must be taken into consideration when the wastes are used as building materials [34,35]. The leached out heavy metal contents should be low enough to meet the requirements [36,37]. Barna et al. reported leaching assessment of road materials containing primary lead and zinc slags. They found that the release of Pb and Zn was controlled by the pH of the leachate [38]. Zhou et al. evaluated the toxicity risks of lead-zinc sulfide tailing-based construction materials with sulfuric acid and nitric acid method, by which the specimens were leached in a mixed acid solution with a pH of 3.20 ± 0.05 [39]. Onuaguluchi and Eren reported the toxic metal immobilization properties of copper tailings as a potential additive in concrete. The leaching of heavy metals was evaluated by toxicity characteristic leaching procedure with leaching medium of glacial acetic acid and deionized water [40]. In order to better understand the leaching performance of tailing concrete, the leaching medium should cover different acid-based conditions.

Science coarse aggregate (CA) is the largest component of high strength concrete, significantly affecting the performance, it is necessary to investigate the characteristic of ATCA in tailing concrete. Furthermore, the toxicity risks of ATCA concrete should be evaluated to ensure the concrete is harmless to the environment. The aim of this study is to investigate the characteristic of ATCA and the leaching performance of ATCA concrete based on the previous work. A series of experiments was performed, such as chemical composition and mechanical properties of ATWRs, coarse quality of ATCA, and the toxicity risks of ATCA concrete, which were tested and discussed in the present work.

The rest of this paper is organized as follows. Section 2 introduces the background of a real case study. Section 3 describes the main chemical composition of ATCA. Section 4 illustrates the mechanical properties of ATWR and compared with natural rock (NR). Section 5 shows the quality of ATCA and compared with NCA. Section 6 discusses the toxicity risks of ATCA concrete. Finally, Section 6 concludes the article and gives some recommendations for future work.

## 2. Background of a Real Case Study

The construction of the Long-lang highway project, which is nearby XKS and located in South China’s Loudi City, Hunan Province, is chosen as a real case in the present work. As illustrated in Figure 2, the project is 74 km large, including 14 bridges with a total length of 5.1 km. All of the girders of the superstructures of the bridges use C50 concrete, which consumes about 1,330,000 m^3^ of NCA. However, the NCA of the nearby quarry is not suitable for C50 concrete, and the qualified supplier is 52 km to the site. After our investigation, we found that the compressive strength of the ATWRs from XKS met the requirements. Moreover, the distance from XKS to the project is only 20 km. Considering the above issues, the entire C50 concrete is intended to prepare with ATCA as a complete replacement of NCA. Using ATCA can not only provide a new method for the treatment of antimony tailing, but also solve the problem of shortage of raw materials for the project and create huge economic benefits.

In this part, the investment and environmental benefits are discussed by using ATCA concrete. The cost for ATCA administration, reclaiming, and transportation are respectively saving 90% (¥0.8 yuan, Chinese currency), 85% (¥27 yuan), and 61% (¥25 yuan) per cubic meter compared with NCA. The cost saving of direct investment resulting from ATCA can be obtained with the following equation,
(1)$$\Delta C_{AT}=V_{AT}\times (\Delta A_{AT}+\Delta R_{AT}+\Delta T_{AT})$$
where $\Delta C_{AT}$ (yuan) represents the cost saving of direct investment, $V_{AT}$ is the ATCA total volume consumption, $\Delta A_{AT}$ (yuan), $\Delta R_{AT}$ (yuan), and $\Delta T_{AT}$ (yuan) are respectively the saving cost for ATCA administration, reclaiming, and transportation. Based on the investigation, it is conservatively estimated that the ATCA can save 79% of the cost (¥70.224 million yuan, $10.871 million dollars, the exchange rate between US dollar and Chinese RMB is 6.46 on 8 September 2021).

In addition, the use of ATCA reduces about 15,000 m^2^ land occupation for stacking ATWRs. It is very conducive to control the solid waste pollution on the environment and protect the ecology. Therefore, the utilization of ATCA as a complete substitute for NCA in concrete shows a great potential for economic and environmental benefits.

## 3. The Main Chemical Composition of ATCA

### 3.1. Sampling

The test sample was obtained by the five-point sampling method in the antimony waste ore stacking area in XKS. The stacking area is approximately a rectangle, divided into four areas by diagonals, and the sampling points are distributed at the intersection and quarter points of the diagonals. A total of 400 kg ATWRs were collected from each sampling point, 1 m from the upper surface. The collected ATWRs were mixed together and brought back to the laboratory. Some were processed into specimens for rock mechanical performance testing. The rest were crushed into ATCA, of which the particle size was 5–20 mm, used for chemical composition analysis, aggregate quality testing, and preparing for ATCA concrete performance experiment.

### 3.2. Experiment Methodology

X-ray fluorescence spectrometer (XRD, D8 Advance, BRUKER AXS GMBH, Karlsruhe, Germany) was used to analyze the chemical composition ATCA. Polarizing microscope (Leica DM2500P, Leica instruments (China) Co., Ltd., Beijing, China) was used to analyze the mineral composition of the test sample.

### 3.3. Results and Discussion

The chemical composites of ATCA is provided in Table 1. Polarizing microscope and X-ray diffraction (XRD) analysis illustrated that the essential mineral was quartz (SiO_2_), and the subordinate mineral contained kaolinite, pyrophyllite, muscovite, and montmorillonite (Figure 3). It is shown that ATWR is a massive block structure and is mainly composed of quartz.

Figure 4 illustrates mineral composition of ATWR. According to the crystal size and the mosaic relationship between each other, the quartz in the samples can be divided into early and late stage, and the early stage is the main one. The early stage of quartz (Q_1_) is closely inlaid to form a dense aggregate with clear boundaries. Most of the grains are concavo-convex and suture-shaped. The crystal size is relatively uniform, generally varying from 20 μm to 150 μm. A small amount of chalcedony is distributed between grains. The late stage of quartz (Q_2_) is composed of fine vein-like aggregates, filling along the rock fissures (red dotted area). The vein wall of the late stage of quartz (Q_2_) is relatively straight, and the width of the vein varies from 100 to 300 μm, and the particle size of late stage quartz varies from 30 to 100 μm. The lithology of ATWR is quartzite. According to rock characteristics, it can be inferred that ATWR has good mechanical properties and can be used as CA for high strength concrete.

## 4. Rock Mechanical Properties

### 4.1. Experimental Design

The mechanical properties, such as uniaxial compressive strength, splitting tensile strength, and elastic modulus are the most important geotechnical parameters for intact rocks [41,42]. Since CA is the skeleton of concrete, the mechanical properties of rock affects the mechanical behaviors of concrete. In this paper, uniaxial compressive strength, splitting tensile strength, and elastic modulus are tested according to JTG E41-2005 [43].

To better investigate the rock mechanical properties of ATWR, NR form the qualified supplier for NCA, mentioned in Section 2, was prepared as well. The sizes of the specimens used for the experimental study are listed in Table 2.

### 4.2. Results and Discussions

#### 4.2.1. Compressive Strength of Cubic Samples

The compressive strength of ATWR was 221.51 MPa, while the measured value of NR was 169.52 MPa. Since the lithology of ATWR is quartzite, its compressive strength is much higher than that of NR. There was a clear cracking sound as the ATWR sample was crushed, while the cracking sound of NR was relatively low.

The loading process of both groups of specimens exhibits three stages, i.e., elastic stage, crack developing stage, and failure stage. While the failure modes of the two kinds of rock samples are obviously different, as illustrated in Figure 5. The crushing plane passed along the planar, which could be represented by its inclination angle. According to the experimental data, the angle of ATWR and NR are about 58° and 54°, respectively. Actually, the angle can be derived by the following equation,
(2)$$\beta =\frac{\pi }{4}+\frac{\phi }{2},$$
where *β* is the angle between normal of crushing surface and loading axis; *ϕ* is the friction angle of testing samples, as shown in Figure 5b,d. According to the previous work [44], the higher compressive strength of samples is, the more sliding surfaces. Such phenomenon appears for the strength of ATWR is higher than that of NR.

#### 4.2.2. Splitting Tensile Strength

There are two methods to determine the tensile strength of rock: direct tension test method and indirect tension test method. For the direct tension method, the specimen production is more difficult and the experimental technology is more complicated. Therefore, the indirect tension method is generally used for testing. The failure load of the rock is tested based on the splitting method, and the splitting tensile strength is obtained by conversion.

As shown in Figure 6, splitting tensile strength is tested by a rock mechanics test system (RMT-150C type, Wuhan Institute of geotechnical mechanics, Chinese Academy of Sciences, Wuhan, China). The splitting tensile strength of ATWR is 5.93 MPa, of which 2.7% is compressive strength. While the splitting tensile strength of NR is 2.67 MPa, of which only 1.6% is compressive strength. The splitting tensile strength of ATWR is two times higher than that of NR’s.

#### 4.2.3. Compressive Elastic Modulus

As illustrated in Figure 7, the compressive elastic modulus is tested by the RMT-150C type rock mechanics test system as well. The stress–strain curve is shown in Figure 8. The compressive elastic modulus of ATWR is 3.33 × 10^4^ MPa, while the measured value of NR is 2.38 × 10^4^ MPa. The elastic modulus of NR is 71.5% of ATWR’s. Since compressive strength is an adequate index for mechanical properties, a close relationship exists between compressive strength and elastic modulus of the two kinds of samples.

In this section, the mechanical behaviors such as cubic compressive strength, splitting tensile strength, and compressive elastic modulus of the two kinds of samples were tested simultaneously. The results show that the cubic compressive strength, splitting tensile strength, and compressive elastic modulus of ATWR are 1.31, 2.22, 1.40 times of that of NR’s. Therefore, the mechanical behavior of ATWR is superior to that of NR. This further shows that the ATWR possesses excellent mechanical property for the lithology of is quartzite and that it is very suitable for high strength concrete. It can be inferred that the ATCA as a substitution for NCA possesses perfect physical and mechanical properties. In order to further study the feasibility of ATCA for concrete, the quality of ATCA is investigated in the next section.

## 5. The Quality of ATCA

### 5.1. Experiment Methodology

To ensure the mechanical property [45] and durability [46] of the bridge structure, the quality of concrete materials should be guaranteed. Since the performance of concrete is directly affected by CA, the material quality of ATCA should meet the specification requirements. The ATWRs were sampled and crushed into CA, and the particle size was 5–20 mm. The NCA was obtained from the nearest qualified supplier mentioned in Section 2. The material quality of ATCA and NCA was tested simultaneously according to Chinese National Standard GB/T 14685-2011 [47]. The experiment includes gradation, density, water absorption, alkali silica reaction (ASR), needle-flake content, firmness, crushing value, abrasion resistance, dust content, and clay lump content.

### 5.2. Gradation

The sieving method was used to determine the graduation of ATCA and NCA. The gradation curve is illustrated in Figure 9, which shows that ATCA and NCA distribute within the scope of 5–20 mm. Therefore, the ATCA is continuously graded aggregate, conforming to the requirement of the standard.

### 5.3. Density Water Absorption and ASR

The density and water absorption of CA are key indices of material quality, which indirectly reflects the open porosity of the aggregate. The lower water absorption of the aggregate means lower open porosity, higher density, and better working performance and durability behavior for the concrete.

The apparent density, gross bulk density, loose bulk density, void ratio, and water absorption are listed in Table 3. It is demonstrated that all indices meet the specification requirements. As per the code, if the water absorption is less than 1%, the index is classified as grade Ⅰ, and 1–2% is classified as grade Ⅱ. The results of water absorption of ATCA and NCA are respectively 0.7% and 0.8%, which are both classified as grade Ⅰ. The apparent density, gross bulk density of ATCA are higher than those of NCA’s. While the loose bulk density, void ratio and water absorption of ATCA are lower than those of NCA. Nevertheless, it is explicitly found that the difference in all of the indices between ATCA and NCA are very small. It can be inferred that the influence of ATCA on concrete workability performance, such as slump, apparent density and air content, is negligible.

In order to evaluate the ASR of ATCA, the alkali activity experiment was conducted. ATCA and NCA specimens with dimension of 25 × 25 × 280 mm were prepared for the experiment. For 14 days, all samples were placed in the NaOH solution, of which the concentration was 1 mol/L, and the temperature was (80 ± 2) °C. The length of the sample was measured, by which the expansion rate of length was calculated and the alkali activity of aggregate was evaluated. The expansion rates of ATCA and NCA specimens are 0.71‰ and 0.80‰, respectively, which are both less than 1‰ of the standard required.

### 5.4. Other Indices of Material Quality

Other indices of material quality, including needle-flake content, firmness, crushing value, abrasion resistance, dust content, and clay lump content were test in this subsection. As per the code, all of the indices are classified into three grades, of which the ranges are shown in Table 4.

The content of needle-flake particles was tested by a needle-flake gauge, divided into three grades. As illustrated in Table 4, if the needle-flake content is less than or equal to 5%, it is classified as grade Ⅰ, 5–15% is classified as grade Ⅱ, and 15–25% is classified as grade Ⅲ. The results of tested needle-flake content of ATCA and NCA are respectively 4.5% and 3.5%, which are both classified as grade Ⅰ.

The sample was soaked in a saturated sodium sulfate solution. After five cycles of immersion and drying, the mass loss percentage was measured and the firmness was determined. The results of the mass loss percentage of ATCA and NCA are respectively 3.0% and 3.6%, which are both classified as grade Ⅰ according to Table 4.

The crush value refers to the ability of CA to resist crushing and indirectly reflects the strength performance of CA. Pressure was applied to the sample by a press, and the mass percentage of the sieved fine material after pressure was used to determine the crushing value. The results of mass percentage of the sieved fine material of ATCA and NCA is 9.2% and 14.8%, which are classified as grade Ⅰ and grade Ⅱ, respectively.

The abrasion resistance of CA was tested by the Los Angeles method. The mass loss of ATCA and NCA is 18.0% and 24.5%, respectively. As per the code, the abrasion resistance of ATCA and NCA are both classified as grade Ⅰ.

The dust content was determined by testing the weight of particles with a particle size of less than 0.075 mm in the CA. The results of the dust content of ATCA and NCA is 0.4% and 0.72%, which are classified as grade Ⅰ and grade Ⅱ, respectively.

Moreover, the clay lump content was determined by testing the weight of clay lump with a particle size of more than 2.36 mm in the CA. The results of the clay lump content of ATCA and NCA is 0.1% and 0.18%, which are all classified as grade Ⅱ according to Table 4.

### 5.5. Discussion

The experiment results are summarized and showed in Table 3 and Table 4. It is demonstrated that all of the quality indices of ATCA and NCA meet the requirements of the current industry standards of China.

The results of water absorption and firmness of ATCA are superior to NCA, therefore, it can be inferred that ATCA concrete has better frost resistance. The alkali activity of samples seems increased for ATCA specimens. However, the difference is not significant and can be neglected. The tested results of firmness, crushing value, and abrasion resistance of ATCA are superior to that of NCA. ATCA possesses excellent abrasion resistance performance, thus it can be inferred that ATCA may be applied in concrete pavement. It is demonstrated that the compressive performance of ATCA is significantly better than that of NCA, which is completely consistent with the strength test results of the material. The dust content and clay lump content of ATCA are lower than that of NCA, and it is beneficial to enhance the bonding performance between ATCA and mortar. However, the needle-flake content of ATCA is higher than that of NCA. The factor for higher needle-flake content of ATCA could be attributed to higher compressive strength of ATCA, which tends to produce needle-like particles when crush into CA.

## 6. The Leaching Test of ATCA Concrete

### 6.1. Experiment Methodology

The toxicity characteristic leaching experiment was conducted to evaluate the concentration of heavy metals from ATCA concrete and ensure that the concrete was harmless to the environment. The static soaking method was used to analyze the dissolution and release of the main heavy metals including Sb, As, Hg, Pb, Cd, and Zn in the ATCA concrete under different pH conditions according to GB 30770-2014 [48].

A total of seven batches of leaching tests were carried out to simulate the leaching of heavy metal from ATCA concrete soaked in different pH solutions. Sulfuric acid or sodium hydroxide solution were used to adjust the pH values of the soaking solution at 4, 5, 6, 7, 8, 9, and 10.

Table 5 shows the mixture proportions for the cubic concrete block. Ordinary Portland cement with a strength class of 42.5, ATCA made from the sampled ATWRs, and manufactured sand were used in the experiment. Poly-carboxylate superplasticizer was added to concrete with 0.2% by the weight of cementitious materials.

### 6.2. Results and Discussion

The concentration of the heavy metal was measured every two days and the test lasted for 14 days. The results are shown in Figure 10. It should be noted that only the test results of Sb, As, and Hg are shown in the figure. Since the test results of several other heavy metal elements such as Cd, Pb, and Zn are quite small and lower than 0.001 mg/L, which meet the specification requirements. The results are not presented here.

As can be seen from the figure, pH has a great influence on the leaking of Sb, As, and Hg in the concrete. Figure 10a shows the test result of Sb. When the pH is 4, 7, and 10, the concentration of Sb on the fourth day is respectively 0.247 mg/L, 0.206 mg/L, and 0.245 mg/L; on the eighth day it is respectively 0.266 mg/L, 0.218 mg/L, and 0.265 mg/L; and on the 14th day it is respectively 0.269 mg/L, 0.225 mg/L, and 0.269 mg/L. It can be seen that both acidic and alkaline conditions will promote the dissolution and release of Sb in the concrete. The concentration of Sb increases with time and reaches a steady state on the eighth day. When the pH is 4, 7, and 10, the concentration of Sb on the eighth day is 1.08, 1.06, and 1.08 times of that of the test value on the fourth day.

As for As, the changing trend of the tested result is consistent with Sb, illustrated in Figure 10b. When the pH is 4, 7, and 10, the concentration of As on the fourth day is 0.064 mg/L, 0.039 mg/L, and 0.054 mg/L, respectively; on the eighth day it is 0.067 mg/L, 0.042 mg/L, and 0.056 mg/L, respectively; and on the 14th day it is 0.069 mg/L, 0.045 mg/L, and 0.058 mg/L, respectively. When the pH is 4, 7, and 10, the concentration of As on the eighth day is 1.05, 1.08, and 1.04 times of that of the test value on the fourth day.

While the test results of Hg shows a different vibrational tendency, illustrated in Figure 10c. When the pH is 4, 7, and 10, the concentration of Hg on the fourth day is 0.0027 mg/L, 0.0023 mg/L, and 0.0018 mg/L, respectively; the concentration on the eighth day is 0.0029 mg/L, 0.0023 mg/L, and 0.0018 mg/L, respectively. After the eighth day, there is almost no change in concentration of Hg. It can be seen that the concentration of Hg decreases with pH value increases. Therefore, the acidic condition will promote the dissolution and release of Hg in the concrete, while the alkaline condition will inhibit the leaking of Hg in the concrete.

Under varying conditions of pH range 4–10, the concentrations of Sb, AS, and Hg reach a steady state on the eighth day. Moreover, on the 14th day, the maximum concentrations of Sb, AS, and Hg are 0.266 mg/L, 0.067 mg/L, and 0.029 mg/L, respectively. While the regulatory limits of Sb, AS, and Hg are respectively 0.3 mg/L, 0.1 mg/L, and 0.05 mg/L. It is illustrated that all the test results are less than the regulatory limits. Therefore, the ATCA concrete can be considered as a non-hazardous material.

## 7. Conclusions

The chemical composition and mechanical properties of antimony tailing waste rock (ATWR), quality of antimony tailing coarse aggregate (ATCA), and leaching performance of ATCA concrete are studied in this work. Polarizing microscope and X-ray diffraction (XRD) analysis illustrated that the essential mineral of ATWR was quartz (SiO_2_) and the lithology of ATWR was quartzite. The mechanical behavior such as cubic compressive strength, splitting tensile strength, compressive elastic modulus of ATWR and natural rock (NR) were studied simultaneously. The results show that the cubic compressive strength, splitting tensile strength, and prism compressive elastic modulus of ATWR are 1.31, 2.22, and 1.40 times of that of NR’s, respectively. Therefore, the mechanical behavior of ATWR is superior to that of NR. It is demonstrated that all of the quality control indices of ATCA meet the requirements of the current industry standards of China. Moreover, ATCA possesses excellent abrasion resistance performance, thus it can be inferred that ATCA may be applied in concrete pavement. The leaching of main heavy metals including Sb, As, Hg, Pb, Cd, and Zn in ATCA concrete under different pH conditions are below the regulatory limits. The ATCA concrete can be considered as a non-hazardous material.

A practical case study shows that the use of ATCA can save about 79% of the cost (¥70.224 million yuan, $10.871 million dollars) and reduce about 15,000 m^2^ land occupation for stacking ATWRs. Utilizing ATCA in concrete provides a new way for reusing ATWRs, obtaining significant environmental and economic benefits. It should be noted that there is much future research work to do. The microstructural structure, the bonding with cement paste of ATCA concrete, and the static and fatigue performance of ATCA structure should be investigated.

## Figures and Tables

**Figure 1 materials-14-05480-f001:**
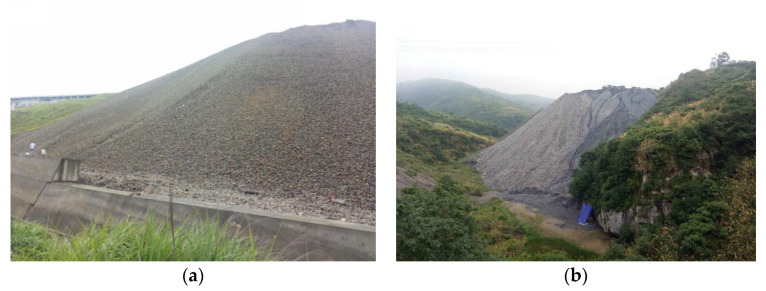
The ATWRs in XKS: (**a**) ATWRs occupy valuable land resource; (**b**) ATWRs destroy the ecological environment.

**Figure 2 materials-14-05480-f002:**
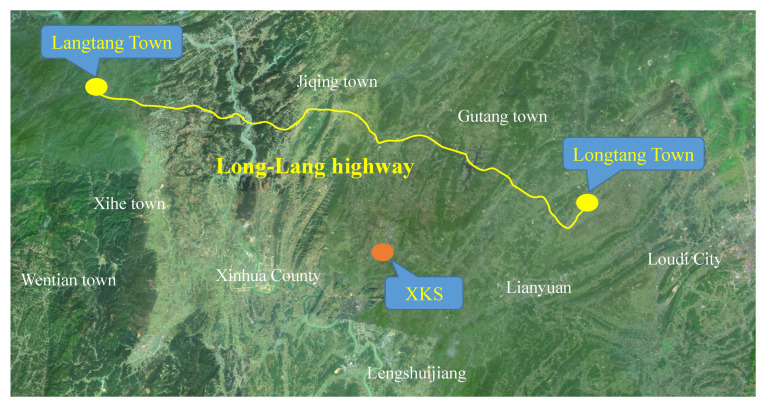
The location of Long-lang highway project and XKS.

**Figure 3 materials-14-05480-f003:**
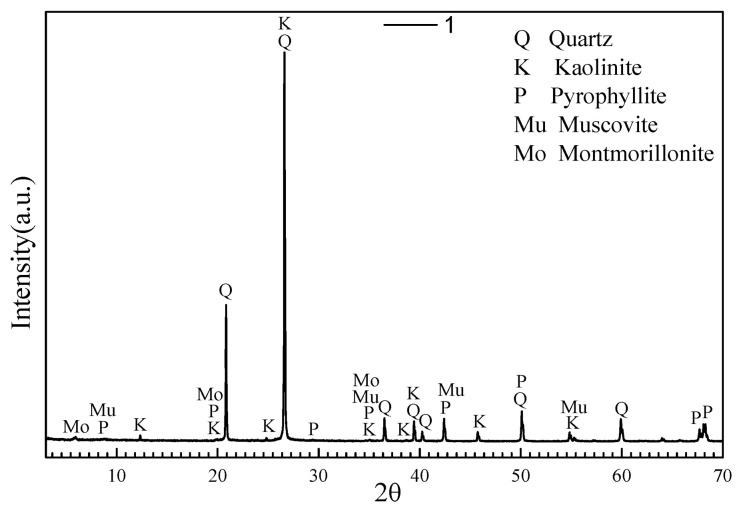
XRD pattern of ATWR.

**Figure 4 materials-14-05480-f004:**
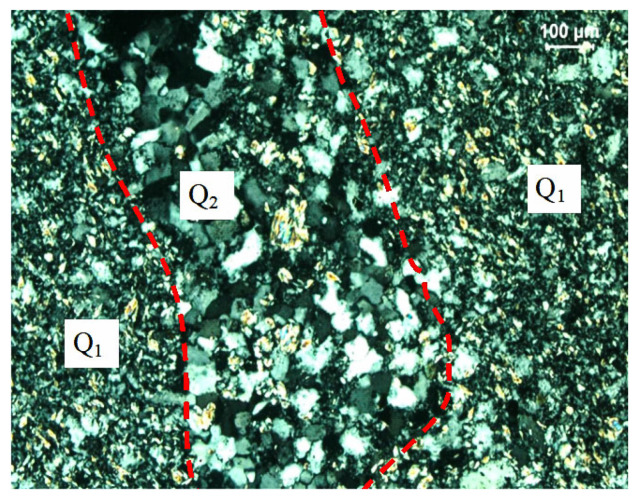
Mineral composition of ATWR. (Q_1_ represents the early stage of quartz, and Q_2_ represents the late stage of quartz).

**Figure 5 materials-14-05480-f005:**
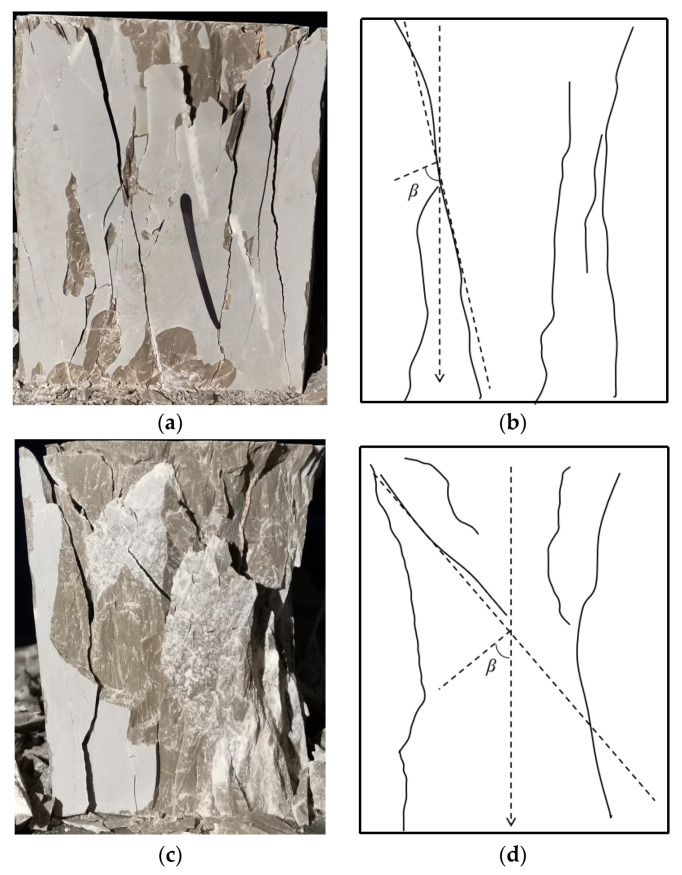
The failure modes of the two kinds rock samples: (**a**) the failure mode of ATWR; (**b**) the crushing plane of ATWR; (**c**) the failure mode of NR; (**d**) the crushing plane of NR.

**Figure 6 materials-14-05480-f006:**
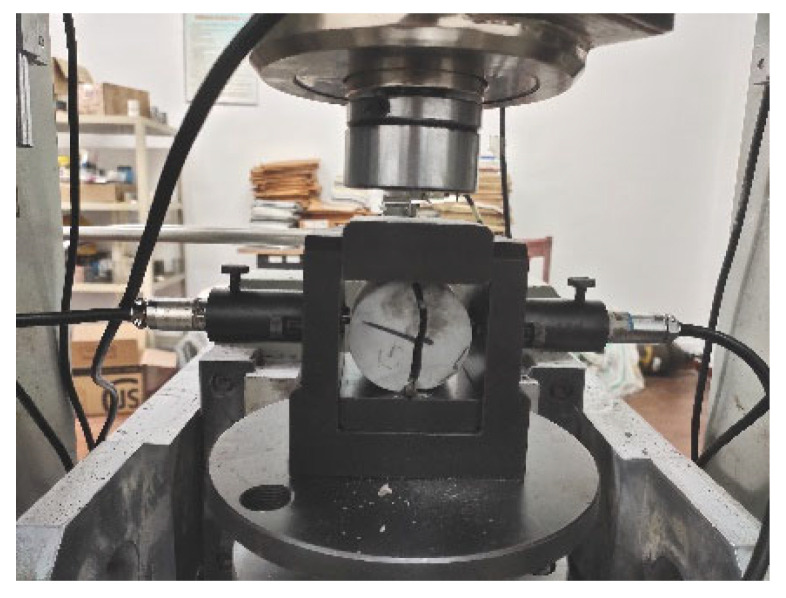
The splitting tensile strength experiment.

**Figure 7 materials-14-05480-f007:**
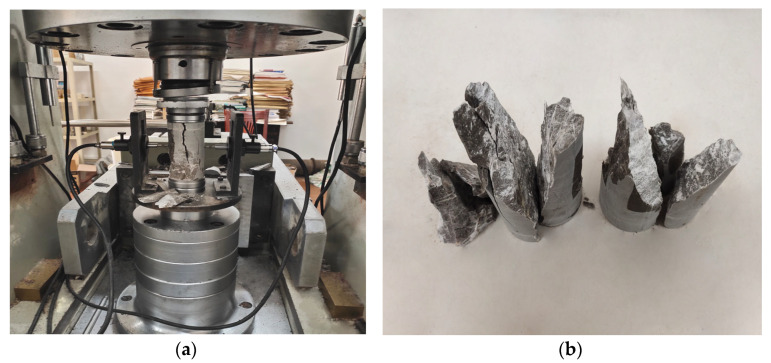
The compressive elastic modulus experiment: (**a**) test instrument; (**b**) failure mode.

**Figure 8 materials-14-05480-f008:**
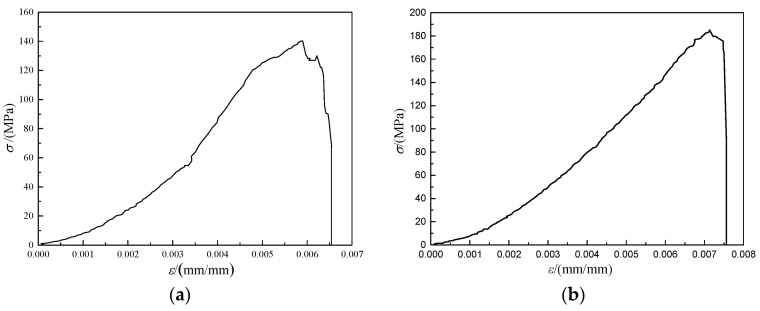
The test result of elastic modulus: (**a**) NR; (**b**) ATWR.

**Figure 9 materials-14-05480-f009:**
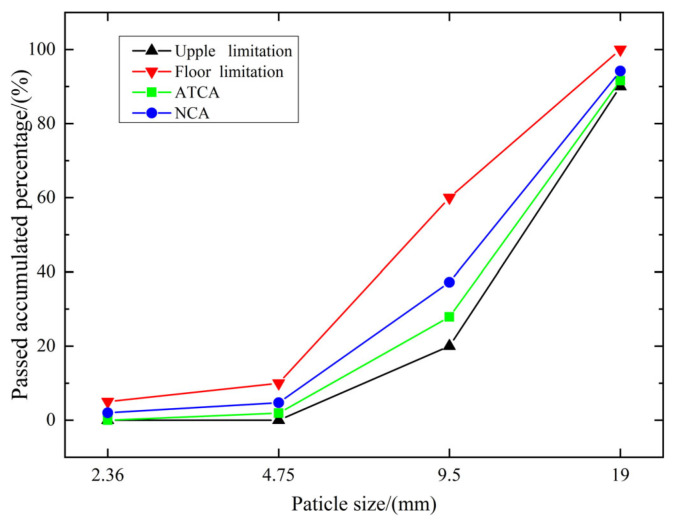
Gradation curve of ATCA and NCA.

**Figure 10 materials-14-05480-f010:**
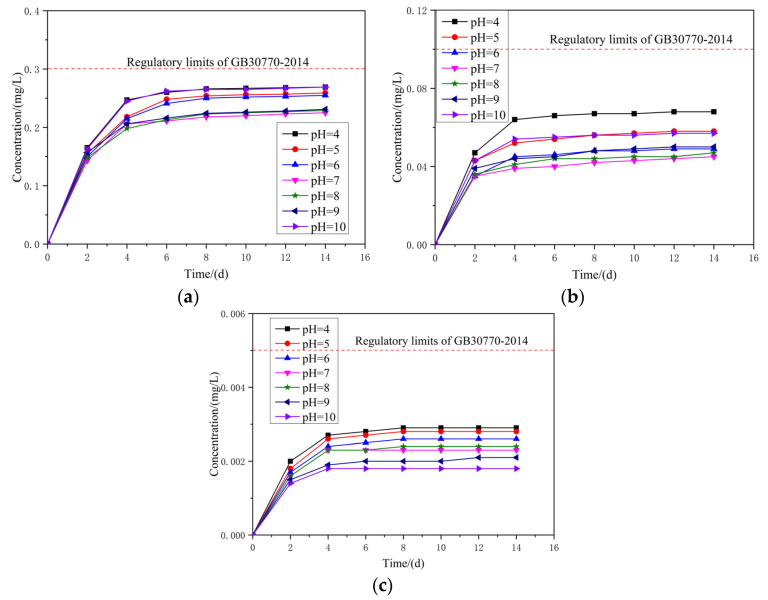
Leaching test result of heavy metals in ATCA concrete (unit: mg/L): (**a**) leaching of Sb; (**b**) leaching of As; (**c**) leaching of Hg.

**Table 1 materials-14-05480-t001:** The test result of main chemical composition of ATCA (%).

Chemical Composition	SiO_2_	Al_2_O_3_	Fe_2_O_3_	CaO	MgO	K_2_O	Na_2_O	TiO_2_	MnO	FeO	P_2_O_5_	LOI
wt.%	88.78	3.86	0.98	0.61	0.11	0.42	0.02	0.17	0.03	0.27	0.04	2.98

**Table 2 materials-14-05480-t002:** Dimensions of the specimens for rock mechanical properties.

No.	Experiments	Sample Shape	Sample Size (mm)
1	Cubic compressive strength	Cube	70 × 70 × 70
2	Splitting tensile strength	Cylinder	Ø50 × 50
3	Prism compressive strength	Cylinder	Ø50 × 100
4	Elasticity modulus	Cylinder	Ø50 × 100

**Table 3 materials-14-05480-t003:** The test results of density, water absorption and ASR.

Description	Apparent Density (kg/cm^3^)	Gross Bulk Density (kg/cm^3^)	Loose Bulk Density (kg/cm^3^)	Void Ratio (%)	Water Absorption (%)	ASR (‰)
ATCA	2653	2592	1497	38.26	0.7	0.80
NCA	2608	2533	1512	38.93	0.8	0.71
Regulatory limits	≥2500	/	≥1350	<47	Grade Ⅰ	<1

**Table 4 materials-14-05480-t004:** The test results of the quality of ATCA and NCA.

Description		Needle-Flake Content		Firmness		Crushing Value		Abrasion Resistance		Dust Content		Clay Lump Content	
		Result	Grade	Result	Grade	Result	Grade	Result	Grade	Result	Grade	Result	Grade
Classification		≤5	Ⅰ	≤5	Ⅰ	≤10	Ⅰ	≤28	Ⅰ	≤0.5%	Ⅰ	0	Ⅰ
		≤15	Ⅱ	≤8	Ⅱ	≤20	Ⅱ	≤30	Ⅱ	≤1.0%	Ⅱ	≤0.5%	Ⅱ
		≤25	Ⅲ	≤12	Ⅲ	≤30	Ⅲ	≤35	Ⅲ	≤1.5%	Ⅲ	≤0.7%	Ⅲ
Test results	ATCA	4.50%	Ⅰ	3.00%	Ⅰ	9.20%	Ⅰ	18.0%	Ⅰ	0.40%	Ⅰ	0.10%	Ⅱ
	NCA	3.50%	Ⅰ	3.60%	Ⅰ	14.80%	Ⅱ	24.5%	Ⅰ	0.72%	Ⅱ	0.18%	Ⅱ

**Table 5 materials-14-05480-t005:** Mix ratio of the concrete (unit: kg/m^3^).

Cement	ATCA	Manufactured Sand	Water	Superplasticizer
485	1047	758	160	5.8

## Data Availability

The data presented in this study are available on request from the corresponding author.

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
