# Peer review of "Sustainable Applications for Utilizing Antimony Tailing Coarse Aggregate (ATCA) in Concrete: Characteristic of ATCA and Toxicity Risks of Concrete"

_materials, 2021, doi:10.3390/ma14195480_

Round 1

Reviewer 1 Report

The authors have experimentally investigated the performance of high strength concrete with complete substitute for natural coarse aggregate with antimony tailing coarse aggregate. The study proves the feasibility of using antimony tailing coarse aggregate high strength concrete. However, this paper is not well-written, and the research is not novel enough to be accepted for publication in Materials. It has already been proved that the antimony waste rock can be reused as a complete substitute of coarse aggregates for high strength concrete. The conclusions drawn in the current paper do not show novel findings compared with previous works by the authors and past research.

See this reference by the authors

Wang, J., Li, B., Li, L., Zuo, C., & Wang, Y. (2021). Feasibility Study of Antimony Waste Rock Coarse Aggregate High Strength Concrete. In Earth and Space 2021 (pp. 123-131).

Other references

Gayana, B. C., & Ram Chandar, K. (2018). Sustainable use of mine waste and tailings with suitable admixture as aggregates in concrete pavements-A review.

Therefore, according to the reviewer's understand and requirement of the journal, the paper is not suitable to be published in its current contents. Please see my comments below:

- The innovation of the research is not clear. The findings in this research are not new and do not add to the current literature.

- Abstract gives information on the main feature of the performed study, but some more details about the obtained results should be added.

-Authors must clarify necessity of this study. Aims and scope should be presented in the last part of introduction. (Research significance in presented, but it is not covered aims and scope of the research).

- The literature review merely lists the published work and does not present the major findings from each effort and how it ties to the presented research. The introduction does not identify the knowledge gap that the authors are trying to address with their research.

- The introduction lists some relevant research but fails to present a scientific review. Please consider to rewrite and clarify the motivation, objectives, and significance of this study.

- The conclusion needs to be refined; it looks like a discussion. Design recommendations and feasibility need to be discussed further.

- The manuscript was not very well prepared. There are a lot of holes. In fact, it reads like a report overall. The presentation is poor, and there lacks of in-depth analysis and discussions. The quality of the figures is low. I cannot recommend it for publication before it is significantly improved.

Author Response

We first would like to express our sincere appreciation to you for your valuable comments on our manuscript. Your valuable comments significantly helped the authors to improve the quality of this manuscript. All the reviewers' comments have been seriously taken into consideration and thoroughly addressed in the revised manuscript, and the explanation/corrections are provided in the following item-to-item response in the attached file. To make the revised manuscript easier for editors and reviewers to read, all changes are printed in blue in the revised manuscript.

Reviewer 2 Report

Please provide properties of antimony tailing coarse 15 aggregates, such as crushing value, flakiness index, abrasion resistance, ASR etc. which are very essential to compare with crushed aggregates and standards.

Microstructural studies are required to see the ITZ development with a new aggregate type that will provide evidence for the bonding with cement paste.

Where is the crushing plane passed at compression testing? It is better to add some evidence to show the fracture plane of both aggregates concretes.

Author Response

(The authors gave the same response as above.)

Reviewer 3 Report

The article is very interesting. It is dedicated to the use of antimony mining waste in the production of concrete. At the same time, both a reduction in the cost of production due to the use of secondary raw materials is achieved, and potentially hazardous waste is disposed of. It should be noted the careful characterization of raw materials, the alignment of road logistics, testing of the obtained concretes not only in terms of physical and mechanical properties, but also toxicity, for which experiments were carried out, as well as the economic calculation of the benefits of the proposed technology. The question is relevant for China with its huge production of antimony and interesting for other countries.

Author Response

(The authors gave the same response as above.)

Reviewer 4 Report

Manuscript is well prepared and globally structure of work is logical. The sections are built somehow different from traditional approach. Section methods is missing and it is incorporated in result subsections for each test. Probably, more traditional structure would give the work more trust (Introduction, materials, methods, results, discussion and conclusions). Economical aspect (background of A Real Case Study) could be represented as separate section. Manuscript can be accepted with some minor revision done.

Other issues regarding to specific paragraphs:

Currency could be expressed as currency or reduction of price in %. Not everyone can calculate benefit in specific currency, and % is better (P3. L.114). “¥0.8, ¥27 and ¥25 per cubic meter than NCA”. Later in the paragraph exchange rate or currency is given, but still relative saving could be used instead (% from original expenditures).

Figure 5 is too general and crushing pattern of cube could be affected by testing surfaces. I would suggest to remove the figure and in discussion analyze common and different observations. Similar with figure 6. It can be removed. Why author put two images for fig.5 (ATWR and NR coarse aggregate), while for Fig.6 only one?

Table 4 needs extension with measured results. Measured results can be put in braces or another column/row. Table 5 somehow duplicates table 4. The title of table 4 can be confusing. Table 4 gives values for CA, not directly for ATWR. This must be checked.

In conclusions only five heavy metals are mentioned (Sb, As, Hg, Pb and Cd), while in results (6.2.) there are six - Cd, Pb, Zn, As, Hg, Pb. The same is in abstract. Was the Zn determined? Correction must be done.

Author Response

(The authors gave the same response as above.)

Round 2

Reviewer 1 Report

The authors have made a considerable effort to revise the manuscript and the main comments about the scientific approach have been successfully addressed. The paper has been significantly improved however, there are several grammatical mistakes throughout the paper and English requires substantial editing to correct these mistakes and improve the quality of the written presentation using appropriate scientific English language. 

Author Response

We would express our sincere appreciation for your carefully review, encouraging recommendation and valuable comments. We have checked in detail and made carefully revise. The grammatical mistakes, spelling errors and the written presentation have been improved. We learned a lot when we revised the paper according to your suggestions. We will pay attention to these issues in our future works. 
